# Genome-wide SNPs reveal novel genetic relationships among Atlantic cod (*Gadus morhua*) from the south coast of Newfoundland, Canada (subdivision 3Ps), Northern cod stock complex, and Gulf of St Lawrence

Sarah Babaei[1,¤,*], Divya A. Varkey[2], Aaron T. Adamack[2], Nathalie M. LeBlanc[1], Gregory N. Puncher[1], Geneviève J. Parent[3], Yanjun Wang[4], Sherrylynn Rowe[5], Cassidy C. D'Aloia[6,‡], Scott A. Pavey[1,‡]

1 Department of Biological Sciences, University of New Brunswick, Saint John, New Brunswick, Canada, 2 Fisheries and Oceans Canada, Northwest Atlantic Fisheries Centre, St. John, Newfoundland and Labrador, Canada, 3 Fisheries and Oceans Canada, Maurice-Lamontagne Institute, Mont-Joli, Quebec, Canada, 4 Fisheries and Oceans Canada, St. Andrews Biological Station, St. Andrews, New Brunswick, Canada, 5 Centre for Fisheries Ecosystems Research, Fisheries and Marine Institute of Memorial University of Newfoundland, St. John, Newfoundland and Labrador, Canada, 6 Department of Biology, University of Toronto Mississauga, Mississauga, Ontario, Canada

¤ Current Address: Department of Ecology and Evolutionary Biology, University of Toronto, Toronto ON, Canada
‡ CCD and SAP are joint senior authors.
* sarah.babaei@unb.ca

## Abstract

The south coast of Newfoundland, Canada (Northwest Atlantic Fisheries Organization (NAFO) Subdivision 3Ps) is known to be a mixing zone for Atlantic cod (*Gadus morhua*). Tagging and genetic studies have shown cod from the Northern and Southern Gulf of St. Lawrence (NAFO Divisions 3Pn, 4RST), Southern Grand Banks (3NO), and the Northern cod stock complex (2J3KL) frequent the waters of 3Ps at various times throughout the year, but the extent of genetic mixing is unknown. However, 3Ps has not been the central focus of previous large-scale genomic analyses of population structure, a knowledge gap that we address using single nucleotide polymorphisms. Using 38,111 neutral markers from reduced representation next-generation sequencing data, we determined the provenance of 3Ps cod relative to the Northern stock complex, Gulf of St. Lawrence, Bay of Fundy, and Gulf of Maine. We present evidence for genetic similarity between 3Ps and the Northern stock complex, particularly NAFO Division 3L. Additionally, genetic clustering analyses suggest 3Ps to be a mixed stock, containing individuals from the Northern stock complex and Gulf of St. Lawrence. Genetic clustering also suggests that there are two subtle subclusters of Northern stock complex and 3Ps cod, indicating there may be subtle population structure within the Northern stock complex and surrounding zones. This new information on population structure gives insight into connectivity and may be useful in future management for rebuilding cod populations.

**Data availability statement:** Raw data is available on Sequence Read Archive under BioProject Accession PRJNA1178142. Minimal datasets are also available on GitHub (https://github.com/SarahBabaei/3Ps_Atlantic_Cod).

**Funding:** Fisheries and Oceans Canada Competitive Science Research Fund grant #2021-22_FS-14_NL to DV Natural Sciences and Engineering Research Council of Canada (NSERC) Discovery Grant to CCD and SAP (RGPIN-2020–04112 & RGPIN-2023-04132, respectively) New Brunswick Innovation Foundation STEM and Social Innovation award to SB.

**Competing interests:** The authors have declared that no competing interests exist.

## Introduction

In recent decades, the field of population genomics has become increasingly integrated into fisheries and wildlife science [1–3]. Next-generation sequencing (NGS) technologies have allowed for the discovery and employment of tens of thousands of single nucleotide polymorphisms (SNPs) to answer questions important for species management and conservation in a timely and cost-effective manner [4]. Alongside ecological research techniques such as mark-recapture and acoustic tagging, population genomics can help us to better understand and identify population structure, delineate population boundaries for conservation, estimate rates of connectivity, and can provide insight into how organisms may adapt to environmental change [2].

Atlantic cod *(Gadus morhua)* is an economically important fish species found in coastal waters on continental shelves throughout the North Atlantic [5]. In Canada, cod are managed using geographic divisions defined by the Northwest Atlantic Fisheries Organization (NAFO, divisions will be referred to using NAFO division names from here on) that extend from off the coast of northern Labrador to southern New Brunswick, including the Gulf of St. Lawrence [6,7] (Fig 1). Cod in some divisions are combined and treated as a single stock, including the Northern stock complex (2J3KL), Northern Gulf of St. Lawrence cod (3Pn, 4RS), and Southern Gulf of St. Lawrence cod (4T-4Vn). In U.S. waters, Atlantic cod are

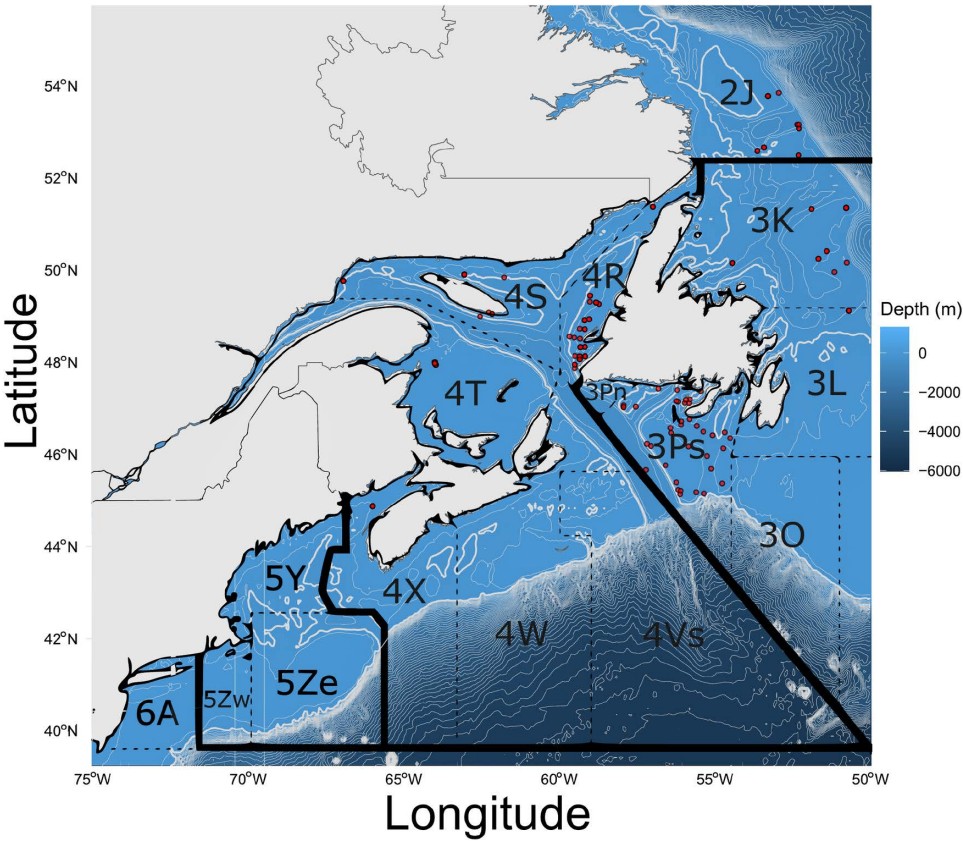

**Fig 1. Sampling locations, including both new 3Ps samples and previously collected samples.** Only points from sets represented in the final, filtered full dataset are plotted in red. Northern stock complex (southern Labrador to Grand Bank; 2J, 3K, 3L), southern Newfoundland (3Ps), Gulf of St. Lawrence (4R, 4S, 4T), Bay of Fundy (4X), Gulf of Maine (5Y).

found in the Gulf of Maine region [8]. Once one of the world's largest marine fisheries, over-exploitation and environmental drivers caused a depletion of this natural resource during the late 1980s and early 1990s [6]. Despite rebuilding efforts, stocks have still not recovered as expected [5,6,9].

Atlantic cod displays chromosomal inversions, a cluster of genes that typically encode for complex phenotypes that are inherited together due to reduced recombination on a chromosome [10–12]. Atlantic cod displays four supergene complexes: Linkage Group (LG) 01, 02, 07, and 12, found on chromosomes 1, 2, 7, and 12, respectively. These inversions contain genes involved in phenotype expression potentially subject to selection from environmental pressures, such as temperature regulation, migratory behaviour, and adaptations to different seawater salinities and oxygen levels [10,13–17]. There is also some evidence from the Northeastern Atlantic that chromosomal inversions may partly influence population structure in Atlantic cod [18,19]. Genomic assessments have identified differences in the genetic profiles of offshore, larger-bodied 'North Sea' and inshore, smaller-bodied 'fjord' ecotypes, and Baltic Sea cod [18,19]. Although the Baltic Sea fish seem to be isolated due to prezygotic reproductive isolation, the North Sea and fjord ecotypes may separate due to behavioural differences (occupation of different depths) or fitness adaptations associated with inversions (e.g., North Sea individuals homozygous for the LG12 inversion display lower fitness in fjords). For Northeast cod off the coasts of Norway, inversions likely influence population structure as well. LG01 harbours 736 genes responsible for swim bladder pressure regulation and skeletal muscle organisation which are important for migratory behaviour that may allow migratory and stationary ecotypes to access different ecological niches. Currently, there is some evidence for linkage group associated ecotypes in Northwest Atlantic cod, but this remains an active area of investigation [20].

Genomics has been used to identify population structure in Northwest Atlantic cod as well [8,13,20–22]. Puncher et al. [8,13] examined population structure and genotypic diversity of offshore samples from the Newfoundland and Labrador Shelf to the Bay of Fundy and Browns Bank using 5,077 neutral SNPs [8,13]. Samples north of 45°N (Northern stock complex and St. Anns Bank) were panmictic and distinguishable from those farther south [8,13] (Bay of Fundy and Browns Bank). A follow-up study identified three genetic clusters: (1) cod north of the Laurentian channel (Bonavista Corridor (3KL), Hawke Channel (2J), Notre Dame Channel (3K), St. Pierre and Green Bank (3Ps)), (2) cod in southwestern Newfoundland waters (Burgeo Bank (3Pn), northeastern Gulf of St. Lawrence (4R), St. Anns Bank (4Vn), and southern Gulf of St. Lawrence (4T)), and (3) cod south of the Laurentian channel [21] (Bay of Fundy, Browns Bank and Southern Scotian Shelf (4X)).

This study mainly focuses on Atlantic cod in the 3Ps subdivision, which has complex population structure and migration patterns. Spawning is known to occur in Placentia Bay, Burgeo Bank, St. Pierre Bank, Halibut Channel, and Hermitage Channel [7,23–29] (Fig 1). Prior to the development of modern genomic tools, mark-recapture sampling was a common method to explore spatial patterns of Atlantic cod movement and remains a widely used tool [30–37]. Tagging experiments have been ongoing since the 1950s and have revealed some connectivity between cod in 3Ps and nearby stocks, such as the Northern cod stock complex (2J3KL), the southern Grand Bank (3NO), and the Gulf of St. Lawrence (4S) [30–38]. For example, Templeman tagged individuals within 3Ps (St. Pierre Bank) and recaptured them during spring and summer on the Grand Bank (3LNO), Halibut Channel, as well as St. Pierre Bank (3Ps) [31]. Recaptures of over thousands of cod within 3Ps showed connections with cod in the Northern Gulf stocks (3Pn4R) and over southern Grand Bank [39,40]. Vertebral counts represent another method explored to delineate cod stock structure [41]; and a mix of types have been found in 3Ps. Overall, results from tagging and other studies suggest

the main areas with linkages to cod in 3Ps were: northeastern Newfoundland and southern Labrador waters (Northern stock complex; 2J3KL), and Southern Grand Bank (3NO) and the Gulf [24–41] (Fig 1).

Though not the main focus, samples from 3Ps have also been included in previous regional genetic studies [20–23,42]. For example, Ruzzante et al. [42] used samples from the Gulf of St. Lawrence (4RT), southern Newfoundland (4Vn, 3Pn, 3Ps), and eastern Scotian Shelf (4Vs) to determine their contributions to overwintering Gulf of St. Lawrence cod. Pre-, post-, and spawning cod were genotyped at six microsatellite loci. They found the highest contributions from waters off southern Gulf of St. Lawrence (NAFO 4T), southeast and central Newfoundland, Cape Breton Island region, and 3Ps bays (Fortune and Placentia Bays), respectively. Genetic distances among groups were also investigated, with the largest distances being between Gulf of St. Lawrence (NAFO 4RT) and southern Newfoundland waters (NAFO 4Vn, 3Pn, 3Ps). This study suggested that changes in population dynamics and structure could be influenced by seasonal differences in spawning and migration. Genetic differences among fish sampled were marginal, perhaps due to low sample size (for example, only three fish were collected from 3Ps) [42].

Northwest Atlantic cod do display differences in spawning times and migratory behaviour. Spawning has been observed in waters off the coast of southern Newfoundland (subdivision 3Ps), including in Placentia and Fortune Bays, Burgeo and St. Pierre Banks, and the Halibut Channel [43]. However, cod do not all spawn in the same season, as there is evidence for Northwestern Atlantic cod spawning from winter to autumn, depending on the region [25,44]. Northern cod stock complex spawn in other areas as well, including the Grand Banks (3L), Belle Isle Banks (3K), and Hamilton Bank (2J) [43–45]. Furthermore, cod are known to undertake long distance migrations [43]. A review of a century of tagging data suggests that individual cod may return to a particular spawning location year after year, a behaviour called 'homing' [43]. Migrating cod move through the Northwest Atlantic during different seasons [43]. Tagging and genetic studies have provided strong evidence that individuals from the northern Gulf of St. Lawrence (4RS) stocks and Northern cod stock complex (3KL) stocks migrate to/through southern Newfoundland waters (3Ps). Cod from farther south (e.g., 4VsWX) and trans-Laurentian cod (4TVn), however, were not seen to move north of the Laurentian Channel [31–33] (including into 3Ps). Thus, tagging and genetic evidence suggests that 3Ps may contain a mixed stock rather than a genetically isolated population.

Building on these studies, it is important to determine the proportion of contribution each of the surrounding populations has to mixed stocks to better inform management and prevent overexploitation of individual populations [8]. Using genomic methods and spatially extensive sampling of juvenile, spawning and non-spawning adults throughout 3Ps, we compare genetic variation in Atlantic cod from 3Ps with that in other populations in the Northwest Atlantic to determine how cod in 3Ps fit relative to these populations.

## Materials and methods

### Sampling

In April and May of 2021 and 2022, a total of 734 scale samples from adults and juvenile cod in 3Ps were obtained from the annual Fisheries and Oceans Canada spring bottom trawl survey (Fig 1; S1 Table in S1 File for individuals included in this study). The survey uses a stratified, random sampling design and is described in greater detail in [46]. Scale samples were stored between paper slips in envelopes at room temperature. Of these 732 samples, we randomly selected 99 samples for inclusion in the sequencing run presented in this study to keep sequencing on one plate.

267 previously collected fin clip and heart muscle samples from juvenile and adult cod were also resequenced for this study, including Northern cod stock complex samples (2J3KL) from spring 2013-2015 published in Puncher et al. [8,13,21] (S1 Table in S1 File) and autumn 2017-2018 samples from annual surveys performed by Fisheries and Oceans Canada and the Atlantic Groundfish Council [8,21] (2017 samples from Northern cod stock are unpublished). These previously collected samples were caught through acoustic-trawl surveys and industry vessels using bottom trawl gear as outlined in Puncher et al. [8,13,21]. Samples were chosen from reference populations (Northern cod stock complex, Gulf of St. Lawrence, Bay of Fundy, and Gulf of Maine) based on DNA availability and quality. Collectively, these sampling locations spanned the southern Gulf of Maine in the U.S. waters and Bay of Fundy to the northern Hawke Channel off Labrador (Fig 1).

### Ethical statement

In Canada, protocols or inclusion in animal use inventories is not required for work involving fishes that are lethally sampled under the Canadian government or other regulatory mandates for established fish inspection procedures, abundance estimates, and other population parameters required for assessing stocks. As the fish collected for this project were sampled as a part of Fisheries and Oceans Canada's annual spring multispecies bottom trawl survey which is used to monitor the abundance of fish and shellfish populations in NAFO Divisions 3LNOPs, these samples fell under that exemption.

### DNA extraction and ddRAD library preparation

DNA from the newly collected 3Ps scale samples was extracted with the Omega E.Z.N.A.® Tissue DNA Kit or Omega E-Z 96™ Tissue DNA Kit (Omega Bio-Tek). The manufacturer's protocol was followed, with modifications when required. For samples with only a small number of scales and/or scales that were too small to reliably remove using forceps, we extracted their DNA from the scales and slime on the paper enclosing the scales following a modified protocol wherein twice the amount of TL buffer was added for lysis. Second, the elution volume and number of elution rounds were modified depending on the sample, with smaller samples having one elution step with 30ul elution buffer and larger samples having one or two elution steps with 100ul total elution buffer. For all samples outside of 3Ps, DNA had previously been extracted.

One next generation sequencing library of 384 individuals was prepared using double digest restriction-site associated DNA sequencing (ddRAD). All individuals were sequenced on one lane to avoid lane effects seen in preliminary results [47]. We used the protocol from Puncher et al. [13], a modified protocol from Poland et al. [48] using enzymes *PstI and MspI,* for consistency with prior ddRAD studies on the reference samples [13,21]. Quality control was done using an Agilent® 2100 BioAnalyzer (Waldbronn, Germany) to confirm fragment sizes of 377-523 prior to paired-end sequencing at 150 bp using an Illumina® NovaSeq™ (San Diego, U.S.A.).

### Bioinformatics and filtering

The STACKS workflow developed by Éric Normandeau (available on github: https://github.com/enormandeau/stacks_workflow) was used to call and filter SNPs post-sequencing based on quality and read coverage. CutAdapt and Process_Radtags STACKS v2.5.2 [49,50] were used to trim adaptor sequences and demultiplex samples, respectively. Sequences were trimmed to 110 bp and aligned to reference genome gadMor3.0 (NCBI accession ID: GCF_902167405.1) using the Burrows-Wheeler Aligner at 90% alignment accuracy (BWA)

[51] Li & Durbin, 2009). *gstacks* from STACKS2 v2.5.2 [40] was used for genotyping. Filtering was done in the *populations* module (removed SNPs with FIS < -0.3, heterozygosity > 0.6) using thresholds based on recommendations from Shafer et al. [52] and previously implemented on cod ddRAD datasets [8,13,21]. VCFtools was used for further filtering, including minor allele frequency (MAF 0.05), minimum read depth (minDP 7), maximum number of alleles (max-alleles 2), maximum percent missing data by locus (max-missing 0.6), maximum percent missing data by individual (max-missing 0.3), and removing related individuals (0.5). Full filtering details are available in S2 Table in S1 File. A Hardy-Weinberg Equilibrium (HWE) filter was not included as literature suggests that HWE filters can obscure subtle population structure commonly seen in marine species [53,54].

To obtain a neutral dataset, the full dataset was further filtered to remove SNPs potentially under selection. This included filtering for linkage disequilibrium (--min-r2 0.2), removing linkage groups, and removing outliers. Linkage groups were removed following methods from Puncher et al. [13]. Pairwise $F_{ST}$ values were calculated for pairs of geographic locations (Northern cod stock complex (2J3KL), Gulf of St. Lawrence (4RST), 3Ps, Bay of Fundy (4X), Gulf of Maine (5Y)) using the R package *pegas* [55]. Manhattan plots were generated for the full dataset (all populations) using these $F_{ST}$ values and the R package ggplot2 v.3.0.0 [56] and used to determine the location of linkage groups. Only loci from the 23 nuclear chromosomes were retained. Post filtering, 52 individuals from 3Ps, 86 individuals from the Northern cod stock complex (29 from 2J, 28 from 3K, 29 from 3L), 136 individuals from Gulf of St. Lawrence cod stocks (64 from 4R, 30 from 4S, 42 from 4T), 12 from the Bay of Fundy (4X), and 33 from the Gulf of Maine (5Y) were retained.

## Outliers

Outliers were detected using both BayeScan v.2.1 [57] (Foll & Gaggioti 2008) and the Bonferroni correction function from the *PCAdapt* R package [58]. Samples were grouped according to their geographic location (Northern cod stock complex (2J3KL), Gulf of St. Lawrence (4RST), southern Newfoundland (3Ps), Bay of Fundy (4X), Gulf of Maine (5Y)). Outliers common between the two methods were removed from downstream analyses using *genepopedit* [59] and compared to the cod genome using ENSEMBL [60]. Gene ontology was investigated using the ENSEMBL and UniProt websites (www.uniprot.org). BayeScan identified 111 outliers, and PCAdapt with the Bonferroni correction identified 25 outliers. All 25 outliers identified by PCAdapt were consensus outliers. To explore the use of a third outlier detection method, we used PCAdapt with a Benjamini-Hochberg correction. This returned 67 outliers but left the consensus outliers unchanged. See S1 Fig in S1 File. for PCA clustering results involving removal of all outliers from all three outlier detection methods. Information for each consensus outlier can be found in S3 Table in S1 File.

## Population structure analyses

To identify spatial patterns of genetic structure, principal component analysis (PCA) and discriminant analysis of principal components (DAPC) were run on all datasets using the *adegenet* package [61]. To further investigate the presence of subpopulation clustering, Bay of Fundy (4X), Gulf of Maine (5Y), and Gulf of St. Lawrence (4RST) were removed to see if the clusters remained. This package was also used for DAPC cross-validation (xvalDapc function, trained using 90% of the dataset with 100 repetitions) to determine the optimal number of PCs to retain, as well as Bayesian and Akaike information criterion plots (find.clusters function with 100,000 iterations). The number of PCs with the consistently lowest root mean squared error (RMSE) was taken, following methods by Jombart and Collins and Deperi

et al. [62,63]. Pairwise $F_{ST}$ and 95% confidence intervals based on 100 bootstrap replications between pairs of populations were determined using the *StAMPP* package in R [64].

## Results

### Sequencing and filtering

Sequencing gave 3,050,194,533 reads. The final datasets had 52 individuals from 3Ps, 86 individuals from the Northern cod stock complex (29 from 2J, 28 from 3K, 29 from 3L), 136 individuals from the Gulf of St. Lawrence stocks (64 from 4R, 30 from 4S, 42 from 4T), 12 from the 4X stock in Bay of Fundy, and 33 from the Gulf of Maine stocks in 5Y. For the 52 3Ps individuals with available maturity data, 53% were immature, 27% were mature but not spawning, 10% were spawning, and 10% were spent. The full dataset had 55,675 SNPs while the neutral dataset had 38,111 SNPs (excluding SNPs in linkage disequilibrium, in chromosomal inversions and outliers). See S1 Table in S1 File for information on each sample and S2 Table in S1 File for details on each SNP filtering step.

### Population structure and connectivity

**Full dataset (including outliers, inversions, and SNPs in linkage disequilibrium).** A PCA with the full dataset revealed three clusters, all of which contained individuals from all populations (S4 Fig in S1 File). These clusters corresponded to LG01 genotypes, as they match those seen in the LG01 PCA [8,21] (S5 Fig in S1 File). A Manhattan Plot using the whole dataset identified four chromosome regions driving most of the differentiation between all population pairs: chromosomes 1, 2, 7, and 12 (Fig 2). Areas of those chromosomes with the highest $F_{ST}$ values match the locations of linkage groups 1, 2, 7 and 12.

**Neutral dataset.** A PCA with 38,111 neutral SNPs and individuals from all sampling sites revealed three genetic clusters. One cluster primarily had individuals from Gulf of Maine (5Y) and Bay of Fundy (4X), one mostly contained individuals from the Gulf of St. Lawrence (4RST), the Northern cod stock complex (2J3KL), and 3Ps, and another contained primarily

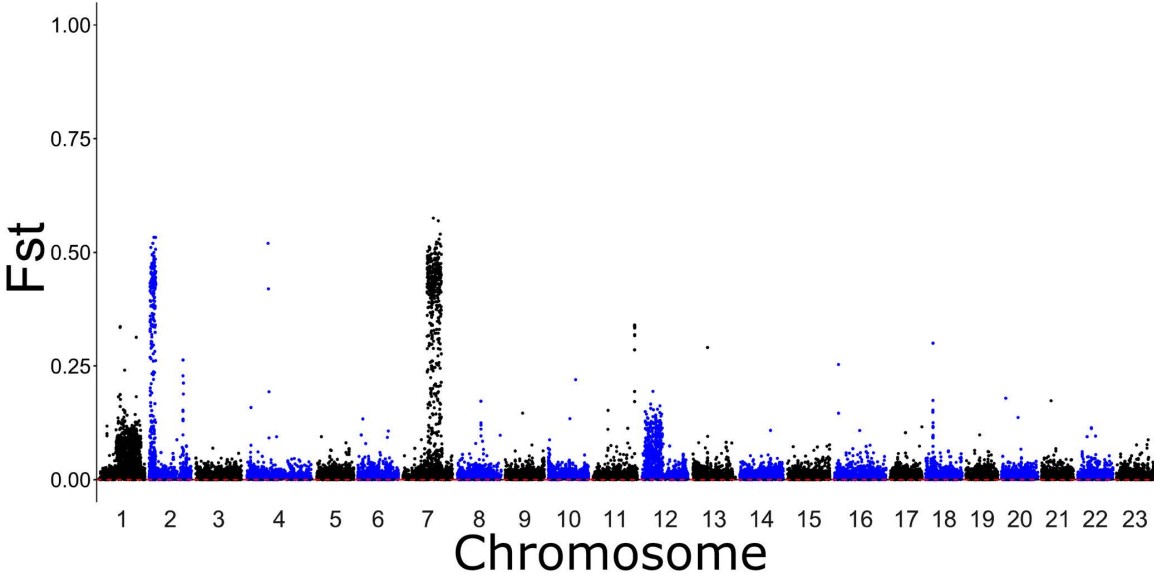

**Fig 2. Manhattan plot using 55,675 SNPs (full dataset).** Highest $F_{ST}$ are seen in parts of chromosomes 1, 2, 7 and 12, corresponding to the inversions.

individuals from the Northern cod stock complex (2J3KL) and 3Ps (Fig 3AB). Results were similar when only juvenile individuals from 3Ps were used (S2 Fig in S1 File). When we removed the divergent cluster comprised of individuals from Gulf of Maine and Bay of Fundy, we found that the two clusters containing Northern stock complex (2J3KL), 3Ps, and Gulf of St. Lawrence (4RST) individuals remained.

As the PCA clusters were close together, i.e., indicating relatively subtle genetic structure, we also ran a DAPC with 50 retained PCs, as suggested by DAPC cross-validation (lowest RMSE of 0.393), to better assign each individual to its cluster (S3 Fig in S1 File). When run with no prior grouping information, the BIC plot displays the lowest BIC at 3 clusters (S6 Fig in S1 File). The DAPC maximised differences between the clusters and showed similar results to the PCA). The frequency of clusters also changes latitudinally. For example, considering the two clusters comprised primarily of Northern Stock / Gulf of St. Lawrence / 3Ps individuals (depicted in gray and white shading in Fig. 4), the proportions shift moving North (3K: 25%/75%, 2J: 14%/86%, Fig 4; Fig 5).

Overall, pairwise $F_{ST}$ values were low (mean = 0.0023), but the relative magnitude of these values corresponded with the PCA/DAPC analyses. The most similar sampling sites were 3Ps and Northern cod stock complex sites in 2J, 3K, and 3L (pairwise $F_{ST}$ = 0.0007), while the most divergent sampling sites were 3Ps and Northern stock complex (2J3KL) with Gulf of Maine (5Y) (pairwise $F_{ST}$ = 0.0044; Table 1).

**Potential genetic substructure within 3Ps/ Northern stock complex/ Gulf of St. Lawrence.** To further investigate the genetic differences between the two PCA clusters

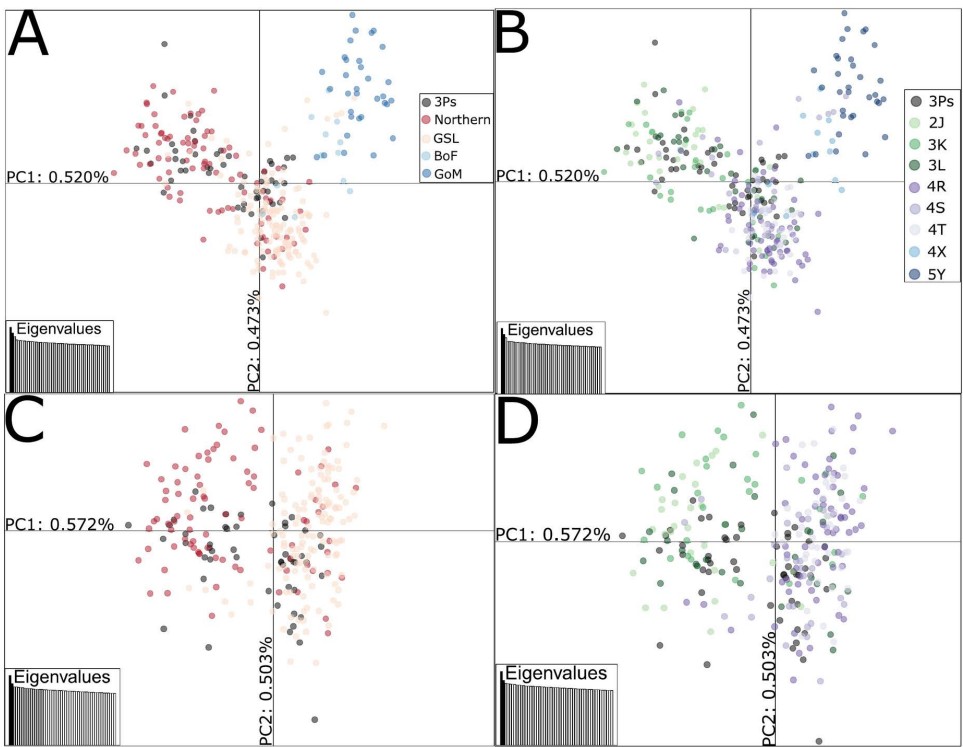

**Fig 3. PCA with neutral dataset, 38,111 SNPs.** A) All geographical locations included, coloured by geographical location B) same as A, coloured by NAFO divisions instead of geographical location C) same legend as A, Gulf of Maine (5Y) and Bay of Fundy (4X) removed D) same legend as B, Gulf of Maine (5Y) and Bay of Fundy (4X) removed. Northern=Northern stock complex (NAFO Divisions 2J3KL), GSL=Gulf of St. Lawrence (NAFO Divisions 4RST), BoF = Bay of Fundy (NAFO Division 4X), GoM = Gulf of Maine (NAFO Division 5Y).

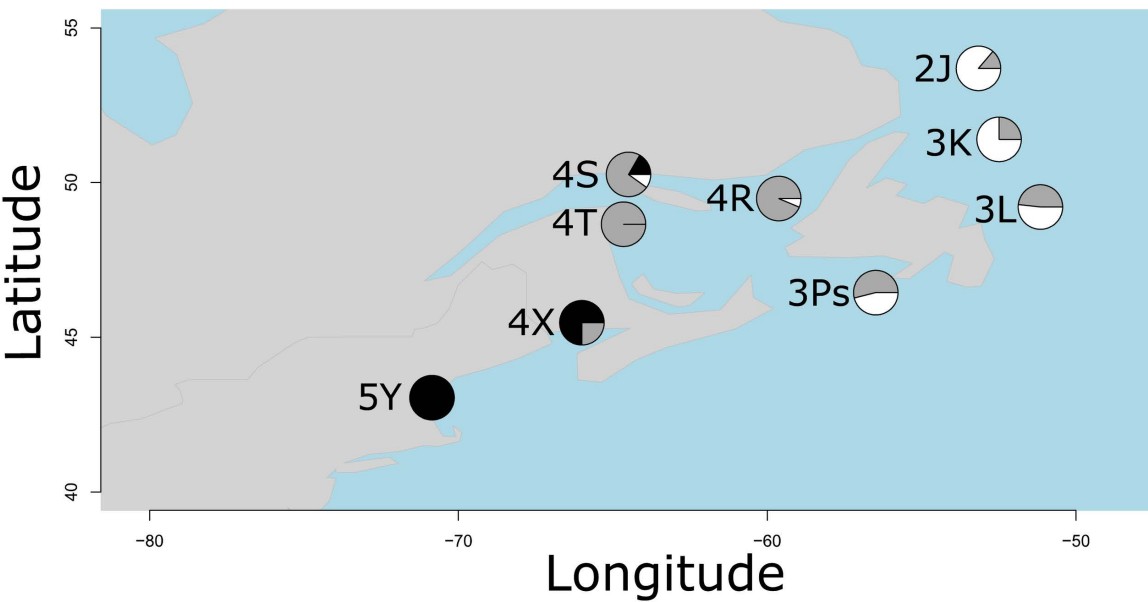

**Fig 4. Proportions of the three groups identified by the discriminant analysis of principal component analysis (DAPC) using 38,111 neutral SNPs.** Northern = Northern stock complex (NAFO Divisions 2J3KL), GSL = Gulf of St. Lawrence (NAFO Divisions 4RST), BoF = Bay of Fundy (NAFO Division 4X), GoM = Gulf of Maine (NAFO Division 5Y).

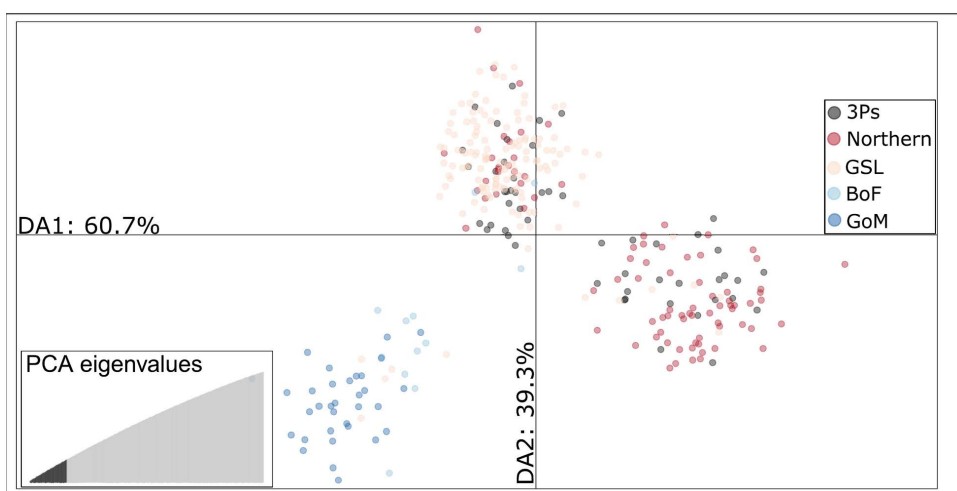

**Fig 5. DAPC using 38,111 neutral SNPs, 50 PCs retained.** Three clusters are seen, one with primarily Gulf of Maine (5Y) and Bay of Fundy (4X) individuals, one with primarily Gulf of St. Lawrence (4RST), Northern cod stock complex (2J3KL), and 3Ps individuals, and another with primarily Northern stock complex (2J3KL) and 3Ps individuals.

containing individuals from 3Ps, the Northern stock complex (2J3KL), and Gulf of St. Lawrence (4RST), we produced corresponding F-statistics. To achieve this, individuals from each cluster were assigned to artificial populations based on whether they clustered in DAPC cluster "1" or "2". These clusters showed a pairwise $F_{ST}$ of approximately 0.0022 (95% CI:0.0021-0.0023). With Gulf of St. Lawrence individuals removed, pairwise $F_{ST}$ was 0.0015 (95% CI:0.0012-0.0018) indicating slightly less differentiation between the clusters with Gulf of St. Lawrence removed.

**Table 1. Pairwise F$_{ST}$ using 38,111 neutral SNPs, outliers removed. * Northern = Northern stock complex (NAFO Divisions 2J3KL), GSL = Gulf of St. Lawrence (NAFO Divisions 4RST), BoF = Bay of Fundy (NAFO Division 4X), GoM = Gulf of Maine (NAFO Division 5Y).**

|  | 3Ps | Northern | GSL | BoF | GoM |
|---|---|---|---|---|---|
| **3Ps** | 0 |  |  |  |  |
| **Northern** | **0.0007** *0.0005–0.0009* | 0 |  |  |  |
| **GSL** | **0.0014** *0.0012–0.0015* | **0.0012** *0.0011–0.0013* | 0 |  |  |
| **BoF** | **0.0033** *0.0027–0.0038* | **0.0017** *0.0012–0.0022* | **0.0017** *0.0012–0.0027* | 0 |  |
| **GoM** | **0.0044** *0.0040–0.0047* | **0.0042** *0.0039–0.0045* | **0.0036** *0.0034–0.0039* | **0.0010** *0.0005–0.0016* | 0 |

\* 95% confidence intervals are presented below the F$_{ST}$ in italics.

We then ran a DAPC without Southern individuals (Bay of Fundy; 4X and Gulf of Maine; 5Y). A DAPC cross-validation suggested 40 PCs for the dataset with Gulf of St. Lawrence individuals (4RST; RMSE of 0.453), and 80 PCs for the dataset without Gulf of St. Lawrence (RMSE of 0.401; S7 Fig in S1 File). Furthermore, using K-means clustering, BIC plots with 40 and 80 PCs retained suggested the presence of 2 clusters with Gulf of Saint Lawrence individuals and 1 cluster without Gulf of St. Lawrence individuals (S7 Fig in S1 File).

To investigate the SNPs most divergent between the two DAPC clusters, a Manhattan plot including and excluding Gulf of St. Lawrence (4RST) was run. This revealed that the SNPs driving the variation between these two clusters were scattered across the genome (Fig 6, S8 Fig in S1 File). Based on the Manhattan plot with Gulf of St. Lawrence, the five SNPs that were most divergent (i.e., displayed highest F$_{ST}$ values) between the two groups were located on chromosomes 18, 21, 13, and 23. One SNP from chromosome 18 drove the most variation between the two groups. This SNP was found inside the gene encoding Neurexin1a (ENSGMOG00000014265), a transmembrane protein that plays a role in anatomical structure development (entry A0A8C5D161_GADMO). The second SNP was in chromosome 21 and was also found between two protein-coding genes: estrogen receptor 2a (*esr2a;* ENSGMOG00000019003) and jagged canonical Notch ligand 2b (*jag2b;* ENSGMOG00000019017). *Esr2a* is located in the nucleus and plays a role in cellular response to estrogen by binding to DNA to allow for transcription (entry A0A8C5FMJ6_GADMO). *Jag2b* is a transmembrane protein that binds calcium ions in the Notch pathway, which plays

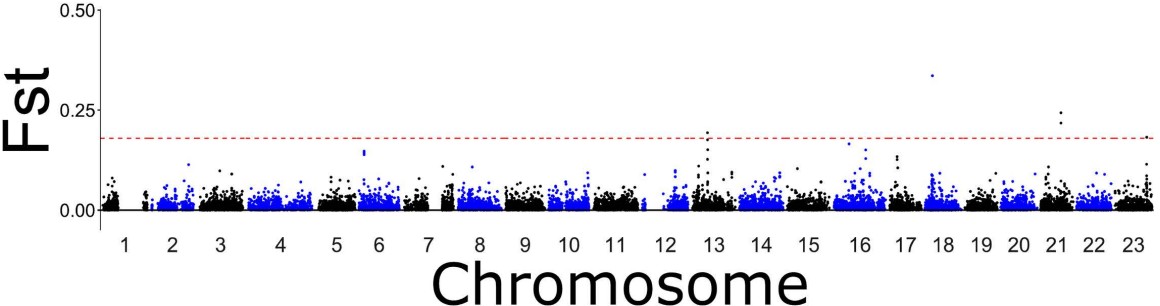

**Fig 6. Manhattan plot showing SNPs with the most variation between the two neutral clusters containing Northern cod stock complex, 3Ps, and Gulf of St. Lawrence.** Gulf of Maine and Bay of Fundy removed. The red dotted line shows the threshold for the 5 SNPs with the highest FST values.

a role in cell and organ development (entry A0A8C5FSJ1_GADMO). The third SNP was found inside *jag2b*. The fourth SNP driving the most variation was located on chromosome 13 and lies next to the R-spondin (*rspo4*; ENSGMOG00000034789) 4 gene and an RNA-coding gene. The *rspo4* protein plays a role in the Wnt signalling pathway and sensory transduction by binding to heparin (entry A0A8C5CBD1_GADMO). Finally, the last SNP was located on chromosome 23 and closest to the bile acid-CoA:amino acid N-acyltransferase-like gene (ENSGMOG00000018681), which plays a role in fatty acid metabolic processes (entry A0A8C4ZRZ9_GADMO). For the top 5 SNPs most divergent between the two groups without Gulf of St. Lawrence, see S4 Table in S1 File.

## Discussion

The 3Ps cod stock, located off the southern coast of Newfoundland, currently has little data available on its genetic composition and population structure. Previous tagging and genetic studies indicate that cod from other regions mix with and move through NAFO Subdivision 3Ps, mainly from the Northern cod stock complex (2J3KL) and northern Gulf of St. Lawrence (4RST) stocks [22,23,31–37]. However, to date, no large-scale genetic studies have been conducted that focus specifically on determining the contributions of each cod population to 3Ps. Using tens of thousands of SNPs throughout the genome, we found that 3Ps cod could not be genetically distinguished from the Northern cod stock complex and that 3Ps cod likely have high rates of gene flow with Gulf of Saint Lawrence cod. Interestingly, we also found that 3Ps and the Northern cod stock complex are potentially comprised of two genetic subgroups.

### Similarities between 3Ps and Northern cod stock complex

Currently, the 3Ps stock is managed separately from other stocks, but cod from the Northern stock complex (2J3KL) are known to move throughout 3Ps [7,8,13,21]. Based on findings from our study, we conclude that some 3Ps and Northern stock complex cod are genetically similar, because there is a similar proportion of the two genetic clusters in both 3Ps and 3L. Juvenile and adult (spawning and non-spawning) cod from 3Ps cluster with Northern stock complex, especially Division 3L, based on both the neutral and full (i.e., including outliers and linkage groups) datasets. Moreover, 3Ps individuals have the most similar genetic profiles to Northern stock complex and traditional F-statistics reveal that there is little difference between 3Ps and Northern stock complex at neutral SNPs.

Consistent with the metapopulation hypothesis [6] we expected some individuals from the Northern cod stock complex to be found in 3Ps since tagging studies recaptured cod tagged in the Northern cod stock complex within 3Ps [31,40,41]. Previous research supporting the metapopulation hypothesis described the Northern stock as composed of multiple genetic populations that mix and interbreed, whereas the isolation hypothesis described the Northern stock complex as consisting of genetically isolated components [65,66]. Thus, our results support the metapopulation hypothesis rather than the isolation hypothesis, even though homing behaviour specific to the 3Ps spawning groups in Placentia Bay has been observed (though we note that we did not have samples from Placentia Bay) [24,43].

### Two potential genetic groups within northern stock complex and 3Ps cod

Interestingly, our results show evidence of complex population structure and connectivity within Northern cod stock complex (2J3KL) and 3Ps cod. Specifically, we show that there are two previously undescribed genetic groups and the genetic relationships in different areas are driven by the relative proportion of these groups. 3Ps and 3L have approximately equal proportions of these groups, whereas 4RST is mostly represented by one and 2J is the other. Thus,

there is a latitudinal gradient in the group proportion for 3Ps and Northern cod stock complex (2J3KL). Preliminary genomic analyses containing more samples from the spring 2015 season show the same patterns [47]. These groups have not been previously described. The clustering also is not associated with inversion genotypes, revealing differences in genome-wide, neutral genetic variation to be the primary driver of the two groups. However, most of the genetic variation is within geographical locations rather than between them, revealing that the potential substructure is subtle, and suggesting that genetic mixing does occur [13,21].

Based on the Manhattan plot using the neutral data set and including Gulf of St. Lawrence individuals (4RST), the five SNPs that display elevated $F_{ST}$ between the two groups are located on chromosomes 18, 21, 13, and 23. The allele frequencies for each of these five SNPs follow a latitudinal pattern, whereby the frequency of these alleles increases either moving north-south or south-north. *Esr2a*, *jag2b*, Neurexin 1a, and *rspo4* all play a role in sexual or structural development, while bile acid-CoA:amino acid N-acyltransferase-like plays a role in metabolism. Clucas et al. [22] also found outliers on chromosomes 18 and 21, including *Esr2*, when comparing cod from St. Pierre Bank (3Ps) to cod in US waters (western Gulf of Maine winter spawners, Georges Bank, Cox Ledge, eastern Gulf of Maine, and Great South Channel). All other outliers they found also corresponded to genes that play a role in various reproductive functions [22]. Due to this, as well as the spatial patterns in these allele frequencies, we speculate that the separation of these clusters may be associated with differences in reproductive function or latitudinal differences in temperature for the northern (Northern cod stock complex Division 2J in particular) and southern stocks (4RST).

## Conclusions, limitations, and future directions

Our study aimed to determine how cod from the mixed 3Ps stock relate to other neighbouring populations and the extent to which other populations contribute to 3Ps. We found 3Ps to be genetically most similar to the Northern cod stock complex (2J3KL). Additionally, we documented previously undetected substructure, where we have found two genetically identifiable groups that extend across the Northern cod stock and subdivision 3Ps cod stock.

To further explore the robustness and composition of these two putative groups, future studies should adopt a more spatially extensive sampling effort that includes more individuals from reference populations throughout the Northwest Atlantic. For example, to determine neutral, fine-scale population structure more clearly within the Northern cod stock complex, more individuals from throughout the Northern stock complex (2J3KL), as well as the southern Grand Banks (3NO) should be included in future analyses. We also note that our study contained more 3Ps juveniles and non-spawning adults than spawning adults, samples with a lack of maturity information, as well as samples from different seasons and years. This combination of samples across maturity stages and time limits our ability to confidently determine each individual's population of origin, as Northwestern Atlantic cod are known to undertake long migrations [65]. Thus, future studies should also aim to include as many spawning individuals as possible to increase confidence in the populations the cod belong to, identify any distinct spawning groups, and consider seasonal variability (e.g., fall versus spring samples).

## Supporting information

**S1 Table. Information for each individual in dataset.**

**S2 Table.**
Filtering information for neutral and full datasets.

**S3 Table. Information for all 25 outlier loci.**

**S4 Table. Top 5 SNPs divergent (highest FSTs) between 2 DAPC clusters in neutral dataset (38,111 SNPs).**

**S1 Fig. PCA with all outliers removed from all three outlier detection methods.**
Detection methods include BayeScan, PCAdapt with Bonferroni correction, and PCAdapt with Benjamini-Hochberg correction. Consensus outliers remained the same with addition of third outlier detection method (25 consensus). Northern = Northern cod stock (2J3KL), GSL = Gulf of St. Lawrence (4RST), BoF = Bay of Fundy (4X), GoM = Gulf of Maine (5Y).

**S2 Fig. PCA with neutral dataset (38,111 SNPs) and only 3Ps juveniles (294 individuals total; 27 3Ps individuals).**
Northern = Northern cod stock (2J3KL), GSL = Gulf of St. Lawrence (4RST), BoF = Bay of Fundy (4X), GoM = Gulf of Maine (5Y).

**S3 Fig. DAPC cross-validation for neutral dataset (38,111 SNPs) with all populations (319 individuals).**
Lowest root mean squared error and highest proportion of successful outcome prediction indicates 50PCs (RMSE of 0.393).

**S4 Fig. PCA with the full dataset of 55,675 SNPs, 319 individuals.**
Three groups are shown along the PC1 axis, corresponding to genotypes from LG01. There is some separation of southern populations along the y-axis (Gulf of Maine, Bay of Fundy) from the from the other populations (Northern, Gulf of St. Lawrence, 3Ps). Northern = Northern cod stock (2J3KL), GSL = Gulf of St. Lawrence (4RST), BoF = Bay of Fundy (4X), GoM = Gulf of Maine (5Y).

**S5 Fig. PCAs for each linkage group.**
Location of linkage groups were determined using the Manhattan plot on the full dataset (see Fig 2). Three clusters are seen in each plot, corresponding to the three possible genotypes. A) Linkage group 1 (LG01; 1892 SNPs). B) Linkage group 2 (LG02; 563 SNPs). C) Linkage group 7 (LG07; 1027 SNPs). D) Linkage group 12 (LG12; 1017 SNPs).

**S6 Fig. Detection of appropriate population clusters in the neutral dataset (38,111 SNPs) with all populations (319 individuals).**
BIC plot with 50PCs retained (as indicated by DAPC cross-validation) BIC plot shows an indication for 3 clusters.

**S7 Fig. DAPC cross-validation plots for 38,111 neutral SNPs.**
A) Including Gulf of St. Lawrence, individuals from Bay of Fundy and Gulf of Maine removed. B) Individuals from Gulf of St. Lawrence, Bay of Fundy and Gulf of Maine removed. BIC plot with 40PCs retained (RMSE of 0.453) C) Neutral dataset including Gulf of St. Lawrence, individuals from Bay of Fundy and Gulf of Maine removed. D) Individuals from Gulf of St. Lawrence, Bay of Fundy and Gulf of Maine removed, 80PCs retained. This is the number of PCs indicated by DAPC cross-validation (RMSE of 0.401). Dip at 2 clusters is seen in only B.

**S8 Fig. Manhattan Plot showing neutral SNPs (38,111 SNPs) with the most variation between the two neutral clusters containing Northern cod and 3Ps.**
Gulf of St. Lawrence, Gulf of Maine, and Bay of Fundy individuals removed. The red dotted line shows the threshold for the top 5 SNPs.

## Acknowledgments

We thank Fisheries and Oceans Canada technicians for collecting scale samples and Genome Quebec for sequencing. We also thank Remy Rochette and Stephen Carr for feedback during SB's MSc thesis defence and two anonymous reviewers for their comments.

## Author contributions

**Conceptualization:** Divya A. Varkey, Aaron T. Adamack, Cassidy C. D'Aloia, Scott A. Pavey.

**Formal analysis:** Sarah Babaei.

**Funding acquisition:** Divya A. Varkey, Aaron T. Adamack, Cassidy C. D'Aloia, Scott A. Pavey.

**Investigation:** Sarah Babaei, Nathalie M. LeBlanc.

**Resources:** Divya A. Varkey, Aaron T. Adamack, Gregory N. Puncher, Geneviève J. Parent, Yanjun Wang, Sherrylynn Rowe, Cassidy C. D'Aloia, Scott A. Pavey.

**Supervision:** Divya A. Varkey, Aaron T. Adamack, Nathalie M. LeBlanc, Cassidy C. D'Aloia, Scott A. Pavey.

**Validation:** Nathalie M. LeBlanc, Gregory N. Puncher.

**Writing – original draft:** Sarah Babaei.

**Writing – review & editing:** Sarah Babaei, Divya A. Varkey, Aaron T. Adamack, Nathalie M. LeBlanc, Gregory N. Puncher, Geneviève J. Parent, Yanjun Wang, Sherrylynn Rowe, Cassidy C. D'Aloia, Scott A. Pavey.

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
