## [Decision Letter · Decision Letter 0]

4 Oct 2024

PONE-D-24-31160Determining population structure of Atlantic cod (Gadus morhua) found along the south coast of Newfoundland, Canada using next-generation sequencingPLOS ONE

Dear Dr. Babaei,

Thank you for submitting your manuscript to PLOS ONE. After careful consideration, we feel that it has merit but does not fully meet PLOS ONE’s publication criteria as it currently stands. Therefore, we invite you to submit a revised version of the manuscript that addresses the points raised during the review process.

Both reviewers raised some important concerns, especially about the outlier analyses that needs to be addressed in the revision.

We look forward to receiving your revised manuscript.

Kind regards,

Sven Winter

Academic Editor

PLOS ONE

Journal Requirements: When submitting your revision, we need you to address these additional requirements. 1. Please ensure that your manuscript meets PLOS ONE's style requirements, including those for file naming. The PLOS ONE style templates can be found at https://journals.plos.org/plosone/s/file?id=wjVg/PLOSOne_formatting_sample_main_body.pdf and https://journals.plos.org/plosone/s/file?id=ba62/PLOSOne_formatting_sample_title_authors_affiliations.pdf 2. Thank you for stating the following financial disclosure: "Fisheries and Oceans Canada Competitive Science Research Fund grant #2021-22_FS-14_NL to DV Natural Sciences and Engineering Research Council of Canada (NSERC) Discovery Grant to CCD and SAP (RGPIN-2020–04112 & RGPIN-2023-04132, respectively) New Brunswick Innovation Foundation STEM and Social Innovation award to SB"  Please state what role the funders took in the study.  If the funders had no role, please state: ""The funders had no role in study design, data collection and analysis, decision to publish, or preparation of the manuscript."" If this statement is not correct you must amend it as needed. Please include this amended Role of Funder statement in your cover letter; we will change the online submission form on your behalf. 3. Thank you for stating the following in the Acknowledgments Section of your manuscript: "This work was funded by Fisheries and Oceans Canada Competitive Science Research Fund grant #2021-22_FS-14_NL to DV and Natural Sciences and Engineering Research Council of Canada (NSERC) Discovery Grant to CCD and SAP (RGPIN-2020–04112 & RGPIN-2023-04132, respectively). We thank Fisheries and Oceans Canada technicians for collecting scale samples and Genome Quebec for sequencing. S.B. was supported by a New Brunswick Innovation Foundation STEM and Social Innovation award. We also thank Remy Rochette and Stephen Carr for feedback during SB’s MSc thesis defence." We note that you have provided funding information that is not currently declared in your Funding Statement. However, funding information should not appear in the Acknowledgments section or other areas of your manuscript. We will only publish funding information present in the Funding Statement section of the online submission form. Please remove any funding-related text from the manuscript and let us know how you would like to update your Funding Statement. Currently, your Funding Statement reads as follows: "Fisheries and Oceans Canada Competitive Science Research Fund grant #2021-22_FS-14_NL to DV Natural Sciences and Engineering Research Council of Canada (NSERC) Discovery Grant to CCD and SAP (RGPIN-2020–04112 & RGPIN-2023-04132, respectively) New Brunswick Innovation Foundation STEM and Social Innovation award to SB" Please include your amended statements within your cover letter; we will change the online submission form on your behalf. 4. When completing the data availability statement of the submission form, you indicated that you will make your data available on acceptance. We strongly recommend all authors decide on a data sharing plan before acceptance, as the process can be lengthy and hold up publication timelines. Please note that, though access restrictions are acceptable now, your entire data will need to be made freely accessible if your manuscript is accepted for publication. This policy applies to all data except where public deposition would breach compliance with the protocol approved by your research ethics board. If you are unable to adhere to our open data policy, please kindly revise your statement to explain your reasoning and we will seek the editor's input on an exemption. Please be assured that, once you have provided your new statement, the assessment of your exemption will not hold up the peer review process. 5. Please include your full ethics statement in the ‘Methods’ section of your manuscript file. In your statement, please include the full name of the IRB or ethics committee who approved or waived your study, as well as whether or not you obtained informed written or verbal consent. If consent was waived for your study, please include this information in your statement as well.

Reviewers' comments:

Reviewer's Responses to Questions

**Comments to the Author**

1. Is the manuscript technically sound, and do the data support the conclusions?

Reviewer #1: Yes

Reviewer #2: Yes

2. Has the statistical analysis been performed appropriately and rigorously? 

Reviewer #1: No

Reviewer #2: Yes

3. Have the authors made all data underlying the findings in their manuscript fully available?

Reviewer #1: No

Reviewer #2: Yes

4. Is the manuscript presented in an intelligible fashion and written in standard English?

Reviewer #1: Yes

Reviewer #2: Yes

5. Review Comments to the Author

Reviewer #1: In this manuscript, Babaei et al looked at the population structure of Northern Cod's west Atlantic populations focusing on individuals from the 3Ps NAFO division. The study uses a reduced representation approach and focuses on neutral loci to estimate population structure. Results suggest that cod in the west Atlantic is a metapopulation and the stock from 3Ps is a mix of Northern Cod (2J3KL) and Gulf of St. Lawrence (3Pn4RS) stocks. The paper is clear and the methodology is appropriate, however, there are major points that should be addressed before I can recommend its publication.

- Although, indeed, previous studies haven't looked specifically at 3Ps, there are population genetics studies in cod that include samples from this NAFO division. Although other papers from the same authors have been cited, I am surprised that these studies haven't been mentioned or discussed in the manuscript: Clucas et al., 2019; Kess et al., 2020; Ruzzante et al., 2000; Therkildsen et al., 2013.

- The analyses are appropriate however, regarding detecting outliers, it is recommended to use at least 3 outliers detection methods as it is well known that different methods detect different sets of outlier SNPs. A more conserved approach would be to consider all SNPs found by BayeScan and PCAdapt as outliers, instead of the common SNPs considering the goal is to have a set of neutral markers. In the manuscript, there is no information on the number of outliers each of the two methods detected, so is unknown the proportion of SNPs that weren’t found in common.

- To confirm that all SNPs selected were neutral, a HWE test should be performed during the filtering step which is not shown in the methods.

- The paragraphs explaining the changes in PCAs when removing genetic groups can be confusing. I recommend using hierarchical genetic structure plots.

- I would recommend carefully reading Clucas et al (2019), they too found outliers in chromosomes 18 and 21, which can enrich the discussion section.

- Do the juveniles cluster differently? Also, it is said that other stocks use the area (3Ps) as spawning ground but not all samples from the other NAFO divisions were sampled during spawning season, this should be discussed.

Other recommendations:

- The title is broad, I would advise a more detailed title.

- I recommend being consistent when using location names vs NAFO divisions, for people unfamiliar with the area can be confusing.

- As seasonality is mentioned, it would be good to include in the introduction a brief description of cod’s migratory behavior and spawning time.

-There are two paragraphs about the inversions in the intro, as the focus of the study is the neutral markers, I recommend reducing this information and providing more details about the genetic structure in the region found by different markers: microsatellites and SNPs array.

Minor comments:

Line 28: Southern should be capitalized?

Line 30: Not true! although not the main focus different previous studies have included 3Ps samples.

Line 38: "Which defines population structure"? not sure what this is referring to.

Line 54: What makes it well-suited for population genomic studies? what species wouldn't be well suited?

Line 61: Not clear the Nov-April?

Line 78: Include reference

Line 120: it is important the sampling of juveniles, spawning and non-spawning adults? nothing in the results or discussion about this.

Line 193: were retained? A verb is missing in this sentence.

Line 198: a more conservative approach is to use all outliers. It is well known that different software detects different sets of outliers. How many outliers were detected by each method?

Line 242: Parts of this should be in the methods.

Line 248: Is exactly half? Would be better to include the percentages.

Line 375: "Another explanation", I am not sure I understand the difference with the explanation in the previous sentence.

Line 378: Complete this paragraph with results from Clucas 2019.

Fig1: Include names of places, people outside Canada might not be familiar with the names.

References

Clucas, G. V., Lou, R. N., Therkildsen, N. O., & Kovach, A. I. (2019). Novel signals of adaptive genetic variation in northwestern Atlantic cod revealed by whole‐genome sequencing. Evolutionary Applications, 12(10), 1971–1987. https://doi.org/10.1111/eva.12861

Kess, T., Bentzen, P., Lehnert, S. J., Sylvester, E. V. A., Lien, S., Kent, M. P., Sinclair‐Waters, M., Morris, C., Wringe, B., Fairweather, R., & Bradbury, I. R. (2020). Modular chromosome rearrangements reveal parallel and nonparallel adaptation in a marine fish. Ecology and Evolution, 10(2), 638–653. https://doi.org/10.1002/ece3.5828

Ruzzante, D. E., Taggart, C. T., Lang, S., & Cook, D. (2000). Mixed-Stock Analysis of Atlantic Cod near the Gulf of St. Lawrence Based on Microsatellite DNA. Ecological Applications, 10(4), 1090–1109. https://doi.org/10.2307/2641019

Therkildsen, N. O., Hemmer-Hansen, J., Als, T. D., Swain, D. P., Morgan, M. J., Trippel, E. A., Palumbi, S. R., Meldrup, D., & Nielsen, E. E. (2013). Microevolution in time and space: SNP analysis of historical DNA reveals dynamic signatures of selection in Atlantic cod. Molecular Ecology, 22(9), 2424–2440. https://doi.org/10.1111/mec.12260

Reviewer #2: General comments:

The study presents compelling results, emphasizing the importance of incorporating genetic analyses into stock management. Overall, the research is robust, but given the range of analyses conducted, it is essential to provide access to all scripts used for data analysis and figure generation. Additionally, I recommend reviewing the names of the locations to ensure consistency throughout the text and figures, as inconsistencies can make the study difficult to follow.

Specific comments:

Introduction

Line 108-111: “Individuals tagged within 3Ps were recaptured during spring and summer on the…”. In the previous sentence, you are referring to several studies and, in this sentence, you mentioned individuals tagged within 3Ps were recaptured but did not say to what study you are referring, which makes the whole paragraph very confusing. In addition, I recommend being more specific instead of using “these studies” in line 109, such as citing them at the end of the sentence.

Materials and Methods

Line 132: “Some previously collected samples” How many previously collected samples were included? Please specify this in the text.

Line 138-140: I would include this information in the previous sentence and make it more objective given you are referring to the same set of “previously collected samples”.

Figure 1 legend: Please add the area that 3Ps represent.

Line 160: Please mention what happened with the other 350 samples that were not included in the library prep.

Line 163: From where did you base the use of these enzymes, did you do any kind of in silico simulation using whole genome?

Line 187: Please include the acronym alongside the names of the geographic locations. These locations are a key element of the study, as the research focuses on stock identification. Ensure the naming is clear and concise to avoid any confusion for the reader.

Line 197: Same thing as the comment before, make sure you are concise with the locations name.

Line 215: What multiple methods?

Results

Sequencing and filtering: Please reference here the Supplementary table 2 so the reader can understand how you ended up with the given number of neutral SNPs.

Line 234-236: Between what populations? This is not clear here nor in the methods.

Line 244: As mentioned before, for the readers that are not familiar with these locations, make sure to include the acronym as well for easier visualization using the map you provided.

Figure 3: Given that your study has multiple locations, you need to make sure the reader does not get confused with the names/acronyms. In figure 3A you used the acronyms and figure 3B the names, which makes extremely confusing to understand and compare the genetic clusters. The same issue is present throughout the text.

Table 2: Are these Fst values significant? Please display the p-value somehow.

Line 291-294: Please also provide p-values for a more comprehensive interpretation of your results, specially considering the fine scale genetic structure you are referring here.

Discussion

Line 375-376: Did the study include perhaps any gene flow or effective migration rate analyses? If not, provide a citation in this sentence that support your affirmation.

Line 378-386: This is a very interesting finding. Is there any reference to support such findings? Perhaps in another species with similar characteristics/distribution?

6. PLOS authors have the option to publish the peer review history of their article (what does this mean? ). If published, this will include your full peer review and any attached files.

**Do you want your identity to be public for this peer review?** For information about this choice, including consent withdrawal, please see our Privacy Policy .

Reviewer #1: No

Reviewer #2: No

---

## [Author Response · Author response to Decision Letter 0]

25 Nov 2024

This response letter has also been uploaded as a separate file along with the revised manuscript as requested by the academic editor.

Academic Editor:

1. Please ensure that your manuscript meets PLOS ONE's style requirements, including those for file naming. The PLOS ONE style templates can be found at https://journals.plos.org/plosone/s/file?id=wjVg/PLOSOne_formatting_sample_main_body.pdf and https://journals.plos.org/plosone/s/file? id=ba62/PLOSOne_formatting_sample_title_authors_affiliations.pdf

>>> The manuscript has been formatted to fit PLOS ONE’s requirements, including all figures.

2. Thank you for stating the following financial disclosure: "Fisheries and Oceans Canada Competitive Science Research Fund grant #2021-22_FS-14_NL to DV Natural Sciences and Engineering Research Council of Canada (NSERC) Discovery Grant to CCD and SAP (RGPIN-2020–04112 & RGPIN-2023-04132, respectively) New Brunswick Innovation Foundation STEM and Social Innovation award to SB"

Please state what role the funders took in the study. If the funders had no role, please state: ""The funders had no role in study design, data collection and analysis, decision to publish, or preparation of the manuscript."" If this statement is not correct you must amend it as needed. Please include this amended Role of Funder statement in your cover letter; we will change the online submission form on your behalf.

>>>We have included a funding statement in the updated cover letter of this manuscript.

3. Thank you for stating the following in the Acknowledgments Section of your manuscript: "This work was funded by Fisheries and Oceans Canada Competitive Science Research Fund grant #2021- 22_FS-14_NL to DV and Natural Sciences and Engineering Research Council of Canada (NSERC) Discovery Grant to CCD and SAP (RGPIN-2020–04112 & RGPIN-2023-04132, respectively). We thank Fisheries and Oceans Canada technicians for collecting scale samples and Genome Quebec for sequencing. S.B. was supported by a New Brunswick Innovation Foundation STEM and Social Innovation award. We also thank Remy Rochette and Stephen Carr for feedback during SB’s MSc thesis defence." We note that you have provided funding information that is not currently declared in your Funding Statement. However, funding information should not appear in the Acknowledgments section or other areas of your manuscript. We will only publish funding information present in the Funding Statement section of the online submission form. Please remove any funding-related text from the manuscript and let us know how you would like to update your Funding Statement. Currently, your Funding Statement reads as follows: "Fisheries and Oceans Canada Competitive Science Research Fund grant #2021-22_FS-14_NL to DV Natural Sciences and Engineering Research Council of Canada (NSERC) Discovery Grant to CCD and SAP (RGPIN-2020–04112 & RGPIN-2023-04132, respectively) New Brunswick Innovation Foundation STEM and Social Innovation award to SB" Please include your amended statements within your cover letter; we will change the online submission form on your behalf.

>>> We have removed the funding information from the Acknowledgments section of the manuscript and added a thank you for the anonymous reviewers. Our funding statement contains all funding information and does not require an update.

>>>All code and genotype data are now available on the paper’s GitHub repository (https://github.com/SarahBabaei/3Ps_Atlantic_Cod). Raw sequencing reads have been uploaded to Sequence Read Archive under BioProject number PRJNA1178142, to be released upon publication.

>>>We have now included an ethics statement in the Methods section of our manuscript.

L181-187: In Canada, protocols or inclusion in animal use inventories is not required for work involving fishes that are lethally sampled under government or other regulatory mandates for established fish inspection procedures, abundance estimates, and other population parameters required for assessing stocks. As the fish collected for this project were sampled as a part of Fisheries and Oceans Canada’s annual spring multispecies bottom trawl survey which is used to monitor the abundance of fish and shellfish populations in NAFO Divisions 3LNOPs, these samples fell under that exemption.

Reviewer 1:

Reviewer #1: In this manuscript, Babaei et al looked at the population structure of Northern Cod's west Atlantic populations focusing on individuals from the 3Ps NAFO division. The study uses a reduced representation approach and focuses on neutral loci to estimate population structure. Results suggest that cod in the west Atlantic is a metapopulation and the stock from 3Ps is a mix of Northern Cod (2J3KL) and Gulf of St. Lawrence (3Pn4RS) stocks. The paper is clear and the methodology is appropriate, however, there are major points that should be addressed before I can recommend its publication.

>>> We are glad the reviewer likes the manuscript overall and would like to thank them for the thoughtful comments which have improved the manuscript. In particular, we have clarified the main take-home message, which is that there are two previously undescribed genetic groups. Two have approximately equal proportions in these groups, whereas 4RST is mostly represented by one and 2J is the other.

- Although, indeed, previous studies haven't looked specifically at 3Ps, there are population genetics studies in cod that include samples from this NAFO division. Although other papers from the same authors have been cited, I am surprised that these studies haven't been mentioned or discussed in the manuscript: Clucas et al., 2019; Kess et al., 2020; Ruzzante et al., 2000; Therkildsen et al., 2013.

>>> We rephrased the text throughout the manuscript to clarify that while other studies have not focused specifically on 3Ps, several have included samples from 3Ps. We now refer to Ruzzante et al. 2000, Clucas et al. 2019, Kess et al. 2020, and Therkildsen et al. 2013 throughout the manuscript.

- The analyses are appropriate however, regarding detecting outliers, it is recommended to use at least 3 outliers detection methods as it is well known that different methods detect different sets of outlier SNPs. A more conserved approach would be to consider all SNPs found by BayeScan and PCAdapt as outliers, instead of the common SNPs considering the goal is to have a set of neutral markers. In the manuscript, there is no information on the number of outliers each of the two methods detected, so is unknown the proportion of SNPs that weren’t found in common.

>>>We agree there are different approaches to identifying and filtering for outliers. Based on your comment, we explored the effect of removing all outliers detected in either method. Their exclusion did not meaningfully change our results, so we left the original analyses in the main text. We also explored a third outlier method (Benjamini-Hochberg) and found that the consensus outliers did not change. When all outliers from all three methods were removed, the results did not substantially change (see figure below). The genepop file with all outliers removed is available on GitHub. Additionally, we now state the number of outliers each method detected in the methods section and added a full list of the outliers to the manuscript’s GitHub repository.

Link to manuscript’s GitHub repository: https://github.com/SarahBabaei/3Ps_Atlantic_Cod

Figure: A) PCA from manuscript B) PCA with all outliers removed from all 3 outlier methods

- To confirm that all SNPs selected were neutral, a HWE test should be performed during the filtering step which is not shown in the methods.

>>> We note that applying a HWE filter to RADseq data can obscure subtle population structure (see Pearman et al. 2022) and thus we intentionally avoided this filter in the original submission. Deviations from HWE (and reductions in heterozygosity) can be caused by many factors beyond selection, with the primary factor of interest Wahlund effects. We have added this to the methods section.

L223-225: A Hardy-Weinberg Equilibrium (HWE) filter was not included as literature suggests that HWE filters can obscure subtle population structure commonly seen in marine species [53, 54].

- The paragraphs explaining the changes in PCAs when removing genetic groups can be confusing. I recommend using hierarchical genetic structure plots.

>>> We substantially revised these paragraphs to better explain the PCA results (L278-283, L289-298). We do conduct a hierarchical analysis insofar as we plot a PCA with (1) all samples (Fig 3A) and (2) the Bay of Fundy and Gulf of Maine removed (Fig 3C). To stay focused on the main results, we removed the supplemental figure that had explored PCAs with various sampling sites removed.

- I would recommend carefully reading Clucas et al (2019), they too found outliers in chromosomes 18 and 21, which can enrich the discussion section.

>>> We thank the reviewer for pointing this out. Based on your comment, we have integrated this paper’s results into the discussion section.

L434-438: Clucas et al. [22] also found outliers on chromosomes 18 and 21, including Esr2, when comparing cod from St. Pierre Bank (3Ps) to cod in US waters (western Gulf of Maine winter spawners, Georges Bank, Cox Ledge, eastern Gulf of Maine, and Great South Channel). All other outliers they found also corresponded to genes that play a role in various reproductive functions [22].

- Do the juveniles cluster differently? Also, it is said that other stocks use the area (3Ps) as spawning ground but not all samples from the other NAFO divisions were sampled during spawning season, this should be discussed.

>>> Based on your comment, we ran a PCA using only 3Ps juveniles and the genetic clusters show almost identical results to the PCA including all individuals (see figure below). Due to the rest of the populations having only 0-5 juveniles and the low sample size of 3Ps (27 juveniles), this figure was not included in the main text of our manuscript. However, we added this figure has to the supplementary information (S1 Figure) and the corresponding dataset has been uploaded to the manuscript’s GitHub repository. Regarding your second point, we added more text about the seasonality of sampling, and the potential implications on our results, to the Discussion.

L293-294: Results were similar when only juvenile individuals from 3Ps were used (S1 Fig.).

L454-459: We also note that our study contained more 3Ps juveniles and non-spawning adults than spawning adults, samples with a lack of maturity information, as well as samples from different seasons and years. This combination of samples across maturity stages and time limits our ability to confidently determine each individual’s population of origin, as Northwestern Atlantic cod are known to undertake long migrations [65].

Link to GitHub: https://github.com/SarahBabaei/3Ps_Atlantic_Cod

Figure: A) PCA from manuscript containing adults and juveniles from 3Ps B) PCA only containing juveniles from 3Ps

Other recommendations:

- The title is broad, I would advise a more detailed title.

>>>Thank you for your comment, we changed the title to be more specific.

Title: ‘Genome-wide SNPs reveal novel genetic relationships among Atlantic cod (Gadus morhua) from the south coast of Newfoundland, Canada (subdivision 3Ps), Northern cod stock complex, and Gulf of St Lawrence.’.

- I recommend being consistent when using location names vs NAFO divisions, for people unfamiliar with the area can be confusing.

>>>We agree and have made changes throughout the MS to improve consistency in naming.

- As seasonality is mentioned, it would be good to include in the introduction a brief description of cod’s migratory behavior and spawning time.

>>>We added a paragraph on cod migratory behaviour and spawning time to the introduction to emphasize the effect seasonality may have on the study.

L129-144: Northwest Atlantic cod also display differences in spawning times and migratory behaviour. Spawning has been observed in waters off the coast of southern Newfoundland (subdivision 3Ps), including in Placentia and Fortune Bays, Burgeo and St. Pierre Banks, and the Halibut Channel [43]. However, cod do not all spawn in the same season, as there is evidence for Northwestern Atlantic cod spawning from winter to autumn, depending on the region [25, 44]. Northern cod stock complex spawn in other areas as well, including the Grand Banks (3L), Belle Isle Banks (3K), and Hamilton Bank (2J) [43-45]. Furthermore, cod are known to undertake long distance migrations [43]. A review of a century of tagging data suggests that individual cod may return to a particular spawning location year after year, a behaviour called ‘homing’ [43]. Migrating cod move through the Northwest Atlantic during different seasons [43]. Tagging and genetic studies have provided strong evidence that individuals from the northern Gulf of St. Lawrence (4RS) stocks and Northern cod stock complex (3KL) stocks migrate to/through southern Newfoundland waters (3Ps). Cod from farther south (e.g. 4VsWX) and trans-Laurentian cod (4TVn), however, were not seen to move north of the Laurentian Channel [31-33] (including into 3Ps). Thus, tagging and genetic evidence suggests that 3Ps may contain a mixed stock rather than a genetically isolated population.

-There are two paragraphs about the inversions in the intro, as the focus of the study is the neutral markers, I recommend reducing this information and providing more details about the genetic structure in the region found by different markers: microsatellites and SNPs array.

>>> We shortened the introductory text about inversions into 1 paragraph (L65-83). We also expanded the introductory text on previous genetic structure assessments of the species in the region, e.g. we added details on Ruzzante et al. 2002.

L116-127: For example, Ruzzante et al. [42] used samples from the Gulf of St. Lawrence (4RT), southern Newfoundland (4Vn, 3Pn, 3Ps), and eastern Scotian Shelf (4Vs) to determine their contributions to overwintering Gulf of St. Lawrence cod. Pre-, post-, and spawning cod were genotyped at six microsatellite loci. They found highest contributions from waters off southern Gulf of St. Lawrence (NAFO 4T), southeast and central Newfoundland, Cape Breton Island region, and 3Ps bays (Fortune and Placentia Bays), respectively. Genetic distances among groups were also investigated, with the largest distances being between Gulf of St. Lawrence (NAFO 4RT) and southern Newfoundland waters (NAFO 4Vn, 3Pn, 3Ps). This study suggested that changes in population dynamics and structure could be influenced by seasonal difference in spawning and migration. Genetic differences among fish sampled were marginal, perhaps due to low sample size (for example, only three fish were collected from 3Ps) [42].

Minor comments:

Line 28: Southern should be capitalized?

>>>Done.

Li

---

## [Decision Letter · Decision Letter 1]

18 Dec 2024

PONE-D-24-31160R1Genome-wide SNPs reveal novel genetic relationships among Atlantic cod (Gadus morhua) from the south coast of Newfoundland, Canada (subdivision 3Ps), Northern cod stock complex, and Gulf of St Lawrence.PLOS ONE

Dear Dr. Babaei,

Thank you for submitting your manuscript to PLOS ONE. After careful consideration, we feel that it has merit but does not fully meet PLOS ONE’s publication criteria as it currently stands. Therefore, we invite you to submit a revised version of the manuscript that addresses the points raised during the review process.

 I agree with the reviewer that the manuscript was greatly improved. I also would agree that if a third outlier analysis was performed, it should be added to the methods and results to show that the outliers are agreed on by various methods. I am confident that if the authors add this final info the manuscript will be suitable for publication.  Please submit your revised manuscript by Feb 01 2025 11:59PM. If you will need more time than this to complete your revisions, please reply to this message or contact the journal office at plosone@plos.org . Please include the following items when submitting your revised manuscript:

We look forward to receiving your revised manuscript.

Kind regards,

Sven Winter

Academic Editor

PLOS ONE

Journal Requirements:

Reviewers' comments:

Reviewer's Responses to Questions

**Comments to the Author**

1. If the authors have adequately addressed your comments raised in a previous round of review and you feel that this manuscript is now acceptable for publication, you may indicate that here to bypass the “Comments to the Author” section, enter your conflict of interest statement in the “Confidential to Editor” section, and submit your "Accept" recommendation.

Reviewer #1: All comments have been addressed

2. Is the manuscript technically sound, and do the data support the conclusions?

Reviewer #1: Yes

3. Has the statistical analysis been performed appropriately and rigorously? 

Reviewer #1: Yes

4. Have the authors made all data underlying the findings in their manuscript fully available?

Reviewer #1: Yes

5. Is the manuscript presented in an intelligible fashion and written in standard English?

Reviewer #1: Yes

6. Review Comments to the Author

Reviewer #1: I reviewed the first version of the manuscript. This version is an improved version and it is clear the authors have made an important effort to attend to both reviewer's comments. I just have one recommendation:

I don't see mention of using the 3rd method (Benjamini-Hochberg) for outlier detection in the methods, I think would be important to add a sentence and add the results removing all outliers in the supplementary material as the authors have already done the analysis.

7. PLOS authors have the option to publish the peer review history of their article (what does this mean? ). If published, this will include your full peer review and any attached files.

**Do you want your identity to be public for this peer review?** For information about this choice, including consent withdrawal, please see our Privacy Policy .

Reviewer #1: No

---

## [Author Response · Author response to Decision Letter 1]

1 Jan 2025

Reviewer #1:

I reviewed the first version of the manuscript. This version is an improved version and it is clear the authors have made an important effort to attend to both reviewer's comments. I just have one recommendation:

I don't see mention of using the 3rd method (Benjamini-Hochberg) for outlier detection in the methods, I think would be important to add a sentence and add the results removing all outliers in the supplementary material as the authors have already done the analysis.

>>>We are glad the reviewer finds this manuscript version improved and we would like to thank them for their recommendation. We have now included a description of the third outlier detection method (Benjamini-Hochberg) in the methods and included the results from removing all outliers from all outlier detection methods in the Supplementary Information (S1 Fig.).

L247-250: To explore the use of a third outlier detection method, we used PCAdapt with a Benjamini-Hochberg correction. This returned 67 outliers but left the consensus outliers unchanged. See S1 Fig. for PCA clustering results involving removal of all outliers from all three outlier detection methods.

L697-701: S1 Figure. PCA with all outliers removed from all three outlier detection methods. Detection methods include BayeScan, PCAdapt with Bonferroni correction, and PCAdapt with Benjamini-Hochberg correction. Consensus outliers remained the same with addition of third outlier detection method (25 consensus). Northern = Northern cod stock (2J3KL), GSL = Gulf of St. Lawrence (4RST), BoF = Bay of Fundy (4X), GoM = Gulf of Maine (5Y).

---

## [Editor Report · Decision Letter 2]

5 Jan 2025

Genome-wide SNPs reveal novel genetic relationships among Atlantic cod (Gadus morhua) from the south coast of Newfoundland, Canada (subdivision 3Ps), Northern cod stock complex, and Gulf of St Lawrence.

PONE-D-24-31160R2

Dear Dr. Babaei,

We’re pleased to inform you that your manuscript has been judged scientifically suitable for publication and will be formally accepted for publication once it meets all outstanding technical requirements.

Kind regards,

Sven Winter

Academic Editor

PLOS ONE
---

## [Editor Report · Acceptance letter]

PONE-D-24-31160R2

PLOS ONE

Dear Dr. Babaei,

I'm pleased to inform you that your manuscript has been deemed suitable for publication in PLOS ONE. Congratulations! Your manuscript is now being handed over to our production team.

Kind regards,

on behalf of

Dr. Sven Winter

Academic Editor

PLOS ONE